# Indoor Experiments on the Moisture Dynamic Response to Wind Velocity for Fuelbeds with Different Degrees of Compactness

**Yunlin Zhang** [1,2]

1 School of Biological Sciences, Guizhou Education University, Gaoxin St. 115, Guiyang 550018, China; zhangyunlin@gznc.edu.cn
2 Key Laboratory of Ecology and Management on Forest Fire in Universities of Guizhou Province, Guizhou Education University, Gaoxin St. 115, Guiyang 550018, China

**Abstract:** The semiphysical method is presently the most widely used for predicting litter moisture content, but it produces some errors. These are mainly due to the simplification of the water loss process and not accounting for the fuelbed structure, which can have a serious impact on the accuracy of litter moisture content predictions and, consequently, on forest fire management. As such, in this study, we constructed fuelbeds with different degrees of compactness, and the moisture content is saturated at this time. The drying process is recorded every 10 min under different wind velocity, and the experiment is stopped when the moisture content is not changing. Taking the saturated fibers' moisture content (30%) as the threshold value, the drying process was artificially divided into two stages (from the initial moisture content to 30%, it is a process of free water drying, and from 30% to the equilibrium moisture content, this is the process of drying of bound water), which is called the distinguishing drying process. The whole drying process (from the initial to the equilibrium moisture content) is called the undistinguishing drying process. Drying coefficient and effect factors were calculated by distinguishing and not distinguishing the drying process, respectively. This established a prediction model based on compactness and wind velocity. The results show that the drying coefficients, $k_2$ and $k$, of the two litter types were significantly different: the $k_2$ of the white oak fuelbed was significantly lower than its $k$, with a maximum variation difference of 57.10%. The $k_2$ in the Masson pine fuelbed was significantly higher than its $k$, with a maximum variation difference of 72.76%. Wind velocity and compactness had significant effects on all the drying coefficients of the two litter types, but with changes in the effect factors. The changes in $k_2$ were weaker than those of the other drying coefficients. Compared with the model that did not distinguish the drying process, the MRE of the prediction models for white oak and Masson pine decreased by 27.39% and 2.35%, respectively. The prediction accuracy of the model of the drying coefficient obtained by distinguishing the drying loss process was higher than that of the model that did not distinguish the drying process. This study was an indoor simulation experiment that elucidated the drying mechanism of litter and established a prediction model for the drying coefficient based on effect factors. It is of great significance for further field verification and for improving the accuracy of moisture content predictions based on the semiphysical method and will significantly improve the accuracy of fire risk and fire behavior prediction.

**Keywords:** wind velocity; compactness; fuelbed; drying coefficient; prediction model

## 1. Introduction

Forest fires first ignite in surface litter, and this litter's moisture content determines the possibility of being ignited and the subsequent series of fire behavior indicators after ignition [1]. Accurately estimating the moisture content of litter is of great significance for fire risk prediction [2–4]. Moisture content obtained using the drying method is the

most accurate, but the subject litter must be dried for at least 24 h, so real-time and future moisture content values cannot be obtained, and the method is difficult to apply in practice. Therefore, the accurate prediction of moisture content and subsequent improvement of the prediction model are the focus of forest fire management research [5].

The semiphysical method is currently the most widely used for predicting litter moisture content. This method takes the water diffusion equation as the main body of the model, in which the parameters are obtained through statistical methods, combining the extrapolation ability of the physical method and the applicability of the statistical method. When the moisture content monitoring step is shortened and the moisture content is less than 35%, the mean absolute error of the prediction is about 0.5–1.3%, and the mean relative error is less than 5% [6,7]. However, when the moisture content exceeds 35% and the fire behavior is predicted based on the moisture content, the accuracy of the resulting predictions sometimes fails to meet the requirements [8,9]. The errors of semiphysical methods arise from insufficient understanding of the dynamic change process of litter moisture content and failure to correct its intermediate parameters (equilibrium moisture content, time lag, and drying coefficient). This is specifically shown in two ways: first, the influence of the fuelbed structure on moisture content change is not considered [10,11], and second, the drying process is simplified [9,12]. For different drying mechanisms, the same key parameters are selected for prediction.

To further improve the accuracy of predictions, scholars have carried out extensive research on the above two problems. According to Matthews et al. [13], compactness is an important feature of the fuelbed structure and has a significant influence on moisture dynamic change, affecting its response to the external environment and the complexity of the moisture diffusion path. Jin et al. quantitatively analyzed the dynamic changes in moisture content in the fuelbeds of Mongolian oak and red pine with different levels of compactness with fixed indoor temperature and humidity [14]. They found that compactness has no significant influence on equilibrium moisture content but had an effect on the drying process. Nelson and Hiers believed that the drying mechanism of the fuelbed is different because the moisture content is divided by the fiber saturation point [15]. When the bed moisture content is higher than the fiber saturation point, it is mainly the free water in the litter that changes through evaporation; conversely, when the moisture content is lower than the fiber saturation point, the drying mechanism becomes diffusion. The drying mechanism and the response of the drying process to the environment are different. Jin and Chen studied the two stages of the drying process in a *Pinus sylverstis* needle bed with the fiber saturation point as the threshold and confirmed that one source of error was choosing the same key parameters for moisture content prediction without distinguishing the drying process [16].

Although extensive studies have been carried out, they are mainly focused on the field monitoring or indoor simulation of temperature and humidity alone in the process of moisture content change, ignoring the effect of wind velocity on dynamic changes in fuelbed moisture content. Currently, there are few studies that separately analyze wind velocity. Jin et al. analyzed the drying process of Mongolian oak broad-leaved beds at different wind velocities and found that wind velocity had no significant impact on the equilibrium moisture content but a significant nonlinear impact on the drying coefficient [14]. Van Wagner analyzed the impact of wind velocity on the drying process of jack pine litter and believed that the impact on the dynamic change in moisture content was different at different time periods and wind velocities [17]. Generally, the litter dried quickly within one hour, and the moisture content change slowed after one hour. Anderson believed that wind velocity has a dual impact on the dynamic change in litter moisture content [18]. On the one hand, it promotes evaporation from the litter through air flow, and on the other hand, it reduces the temperature of the litter, thereby affecting the diffusion of litter moisture content. These studies did not consider the effect of wind velocity on the drying process of fuelbeds with different degrees of compactness, nor did they distinguish the

drying process for analysis. Although the effect of wind velocity is considered in the drying coefficient model of the semiphysical method, the bed structure and drying process are not.

In conclusion, to deeply understand the dynamic change mechanism of the moisture content of litters and improve the accuracy of moisture content predictions based on the semiphysical method, this paper analyzes the response of the drying process of litter to wind velocity at different stages by constructing fuelbeds with different degrees of compactness, simulating different wind velocities indoors, and establishing a prediction model for the drying coefficient based on wind velocity and compactness. As the second largest forest area in China, the forests in southeast China are interlaced with agriculture and forestry, and with high mountains and steep slopes. Once a forest fire occurs, it will have a serious impact on the local ecological environment, people's property, and social stability [19]. The dynamic changes in moisture content in different litter types have different responses to external influencing factors [20,21]; therefore, white oak and Masson pine litter, which are widely distributed and flammable in this region, were selected as the research object.

The purpose of this study is to answer the following questions. (i) Is there a significant difference between distinguishing and not distinguishing the drying process in fuelbeds with different degrees of compactness at different wind velocities? (ii) What are the effects of wind velocity and compactness on evaporation and diffusion in the fuelbed? (iii) Based on compactness and wind velocity, can a prediction model for the drying coefficient considering these two drying processes be established? (iv) Are the drying mechanisms of needleleaf and broadleaf beds the same, and can only one drying coefficient model be used to predict the moisture content of litters as described above? By solving the above problems, a drying coefficient model based on compactness and wind velocity is coupled with the semiphysical method, laying a foundation for subsequent research to improve the accuracy of field moisture content predictions.

## 2. Method

### 2.1. Investigation of Fuelbed Characteristics and Collection of Fuel in the Field

To ensure the practical application of the indoor experiment, this study considers the actual field characteristics of fuelbeds when constructing indoor fuelbeds with different amounts of compactness. Based on this, it is necessary to investigate the outdoor fuelbed and collect litter. The study area is located on Tianhe Mountain (107°43′30″–107°43′38″ E, 27°44′53″–27°45′1″ N), Fenggang County, Guiyang City, Guizhou Province. A standard plot of 20 m × 20 m was established in representative white oak and pine forests. In addition, 30 samples of 20 cm × 20 cm were randomly placed at the standard site to investigate the thickness and compactness of the fuelbed (Table 1). Herein, compactness indicates the porosity (density) of litter in the bed [22], and the calculation formula is shown in Equation (1). The particle density of litter is a fixed value, but the particle density of different litter types is different. Through a literature review and indoor measurements [11], the particle densities of white oak and Masson pine litter were determined to be 543.6 kg·m$^{-3}$ and 623.6 kg·m$^{-3}$, respectively.

$$\beta = \frac{\rho_b}{\rho_p} = \frac{m}{v\rho_p} \tag{1}$$

where $\beta$ indicates the compactness of the fuelbed, $\rho_b$ indicates the bulk density of the fuelbed (kg·m$^{-3}$), $\rho_p$ indicates the particle density (kg·m$^{-3}$), $m$ indicates the quality of the fuelbed (kg), and $v$ indicates the volume of the fuelbed (m$^3$).

**Table 1.** Basic information on the sample and fuelbed characteristics.

| Forest Type | Mean Diameter at Breast Height (cm) | Mean Height (m) | Canopy | Mean Fuelbed Thickness (cm) | Mean Fuelbed Compactness |
|---|---|---|---|---|---|
| White Oak | 16.7 | 14.1 | 0.73 | 4.6 | 0.033 |
| Masson Pine | 22.7 | 16.9 | 0.89 | 3.0 | 0.038 |

The responses of withered and weathered leaves to wind velocity during the drying process are different [18]. In the study area, there was a high annual incidence of forest fires from February to March. Therefore, this study collected weathered litter for indoor simulation experiments to ensure that data on the structure of white oak broad leaves and Masson pine needles were complete and representative. The samples used in this study were collected in February 2022.

### 2.2. Constructing of Indoor Fuelbeds with Different Compactness Levels

According to the investigation of fuelbed characteristics, the mean thicknesses of the white oak broad-leaved bed and the Masson pine needle bed in the field were 4.60 cm and 3.00 cm, respectively. The compactness of the fuelbed of white oak varied from 0.014 to 0.042, with a mean of 0.033. The minimum, mean, and maximum compactness of the fuelbed of Masson pine were 0.016, 0.038, and 0.061, respectively. To ensure practical significance of the study of fuelbeds with similar indoor and field structures, the thickness of the fuelbed of white oak was set at 4.60 cm, and the compactness set to four gradients: 0.014, 0.024, 0.033, and 0.042. The fuelbed thickness of Masson pine was set at 3.00 cm, and the compactness set to five gradients: 0.016, 0.028, 0.038, 0.049, and 0.062.

In this study, litter was installed on an iron frame without a top cover and surrounded by a stainless-steel screen with a mesh diameter of 0.2 mm to maximize the simulation of the effect of outdoor wind velocity on the litter drying process. The length and width of the iron frame were 23 cm and 23 cm, respectively. According to the bed thickness, the bed volumes of white oak and Masson pine were 0.0024 $m^3$ and 0.0016 $m^3$, respectively.

According to Equation (1) and the bed volume set in the laboratory experiment, the litter quantities required for constructing fuelbeds with different degrees of compactness are shown in Table 2. For each compactness, the litter was placed in an oven at 105 °C to dry until the quality did not change. Litter with the corresponding quality at each compactness gradient was taken and completely soaked in water for 24 h. After removal, the litter was placed for a period of time and free water on the surface of the litter was wiped dry with paper towels to construct the fuelbed.

**Table 2.** The quantity of the litter in relation to compactness.

| | White Oak | | | | Masson Pine | | | | |
|---|---|---|---|---|---|---|---|---|---|
| Compactness | 0.014 | 0.024 | 0.033 | 0.042 | 0.016 | 0.028 | 0.038 | 0.049 | 0.062 |
| Quantity (g) | 20.73 | 35.80 | 49.62 | 63.98 | 13.52 | 23.34 | 32.36 | 41.73 | 52.37 |

### 2.3. Drying Experiment at Different Wind Velocities

The wind velocity in the forest is generally less than 5 m·s$^{-1}$ [23], so five values of wind velocity were considered: 1 m·s$^{-1}$, 2 m·s$^{-1}$, 3 m·s$^{-1}$, 4 m·s$^{-1}$, and 5 m·s$^{-1}$. A fan was selected as the source of wind, a handheld weather station (Kestrel nk4500) was used to measure the wind velocity at the central position of the fuelbed, and the selected wind velocity requirements were obtained by adjusting the distance between the fan and the fuelbed. The fuelbed was weighed every 10 min until the quality change in the fuelbed was less than 1%. The air temperature and relative humidity were recorded at each weighing. Experiments were conducted 3 times for each wind speed and compactness, with a total of 20 ratios of white oak conducted 60 times, and a total of 25 ratios of pine conducted 75 times.

From the initial moisture content to the time when the moisture content does not change (equilibrium moisture content), it is called the drying process. At this time, it is called the undistinguishing drying process. Taking the saturated moisture content of fiber (30%) as the threshold value, the drying process is divided into two stages: (1) from the initial moisture content to 30%, which is a process of free water drying; (2) decreases from 30% to the equilibrium moisture content, which is the process of drying of bound water. The combination of these two processes is called the distinguishing drying process.

### 3. Data Analysis

*3.1. Drying Curve*

We calculate the moisture content of the fuelbed on a dry-mass basis, taking time as the horizontal coordinate and the moisture content of the fuelbed as the vertical coordinate. Drying curves were constructed for white oak and Masson pine fuelbeds with different degrees of compactness and wind velocity conditions.

*3.2. Basic Principle of the Drying Coefficient of the Fuelbed*

Byram showed that when the air temperature and relative humidity were fixed, a dynamic change in the moisture content of the fuelbed existed, as shown in Equation (2) [24].

$$\frac{\partial M}{\partial t} = -\frac{(M - E)}{\tau} \tag{2}$$

where $M$ and $E$ indicate the moisture content and equilibrium moisture content at time $t$, respectively, $\tau$ indicates the time lag (h), and $t$ indicates the time (h).

In Equation (2), $\frac{\partial M}{\partial t}$ indicates the drying rate of the fuelbed. It can be seen from Equation (2) that the drying rate is not constant throughout the drying process due to the effects of factors such as current moisture content, equilibrium moisture content, and time lag of the fuelbed. It is difficult to analyze the effect of wind velocity and bed structure. Time lag is only related to fuel characteristics and bed structure; therefore, it is often used to characterize the speed of drying rather than the rate.

Wind velocity affects the drying process of the fuelbed: when there is a certain wind velocity, the dynamic change in the moisture content of the fuelbed can be expressed as: $\frac{\partial M}{\partial t} = -\lambda \frac{(M-E)}{\tau}$ (where $\lambda$ indicates the effect coefficient of wind velocity on drying rate and is dimensionless. When there is no wind, $\lambda = 1$.)

Let $k = \frac{\lambda}{\tau}$, which denotes the drying coefficient, represent the speed of drying of the fuelbed. The larger the drying coefficient, the faster the moisture content loss of the fuelbed, and Equation (2) can be changed as shown in Equation (3). When the air temperature and humidity are relatively stable, the drying coefficient is only related to the wind velocity and bed characteristics, which can better reflect the effect of wind velocity than the specific moisture loss rate. Therefore, it is generally better to select the drying coefficient $k$ to analyze the effect of wind velocity on the drying process of fuelbeds with different bed structures.

$$\frac{\partial M}{\partial t} = -k(M - E) \tag{3}$$

Air temperature and relative humidity have significant effects on the drying coefficient [25,26]. The mean temperature variation of the experiments in this study was 2.3 °C, and the mean relative humidity variation was 1.6%, which could be considered approximately constant (Jin et al., 2016). Therefore, it can be assumed that if the effects of wind velocity and fuelbed structure are not considered, the drying coefficient of each fuelbed is almost constant.

The Simard model is the simplest widely used prediction model among the four calculation methods of equilibrium moisture content [17,27–29]. It is also the calculation method used in the national fire risk rating system of the United States. It has high accuracy in predicting the equilibrium moisture content of the fuelbed (the mean absolute error obtained by Zhang et al. using this method is only 0.64%, and the mean relative error is 3.47% [30]). In addition, since this method does not need to re-estimate parameters, there is only one parameter to be estimated: the drying coefficient; this avoids the uncertainty caused by the dependence of the drying coefficient on the equilibrium moisture content

parameter. Therefore, the Simard model was chosen to calculate the equilibrium moisture content of the fuelbed in this study. The model form is shown in Equation (4).

$$E = \begin{cases} 0.03 + 0.2626H - 0.00104HT & H < 10\% \\ 1.76 + 0.1601H - 0.0266T & 10\% \le H < 50\% \\ 21.06 - 0.4944H + 0.005565H^2 - 0.00063HT & 50\% \le H \end{cases} \quad (4)$$

where $E$ indicates the equilibrium moisture content (%); $T$ indicates the air temperature (°C); and $H$ indicates the relative humidity (%).

### 3.3. Calculation of the Drying Coefficient of the Fuelbed

When the drying process is not distinguished, there is only one stage, so the drying coefficient is only one. When distinguishing the drying process, there are two drying coefficients: from the initial to 30% and from 30% to the equilibrium moisture content, so two drying coefficients need to be calculated. The method of all drying coefficients is the same, using Equations (3) and (4).

If the drying process is not distinguished, for each fuelbed drying experiment under each ratio, the calculation should be made from the second weighing to the last weighing (n th). In Equation (3), $\partial t = 10$ min $= 0.167$ h, $\partial M = M_2 - M_1$ (where $M_1$ and $M_2$ are the previous and current moisture content, respectively), and $M = M_1$. The equilibrium moisture content ($E$) was calculated using the Simard model, and $E = E_1$. At this time, there was only one parameter (drying coefficient $k$) to be estimated in the model. MATLAB (2021) was used for nonlinear parameter estimation, and $k$ was calculated. The arithmetic mean of the $k$ values of three repeated tests was taken as the final $k$ value under this condition.

If the drying process is distinguished it is necessary first to determine the appropriate cutoff point. The mechanisms of different processes are different. According to the drying curves of the white oak and Masson pine fuelbeds, the drying process can be approximately regarded as a linear decline from the initial moisture content to 30%, but once the moisture content is lower than 30%, the drying process tends to gradually slow down. Combined with the earlier work of Luke and McArthur [31], in this study, 30% was taken as the cutoff point, and the process of drying was divided into two stages. The method of calculating the $k$ value in the two stages of the drying process was the same as that in the case of no distinction of the drying process. Among them, the drying coefficient in the first stage was calculated from the initial moisture content to the threshold value (30%), which is denoted as $k_1$. The drying coefficient of the second stage was calculated from the time when the moisture content of the fuel bed was lower than 30% to the last moisture content and is denoted as $k_2$.

### 3.4. t Test

A one-way analysis of variance was used to compare whether there was a significant difference in the drying coefficient between the distinguishing and undistinguishing drying processes.

### 3.5. Variance Analysis

With the three drying coefficients as dependent variables and fuelbed compactness and wind velocity as independent variables, variance analysis was carried out to obtain the factors that had a significant effect (significance was defined as a $p$ value of less than 0.05) on the drying coefficient. An effect diagram and a trend line of the effect of the factor on the drying coefficient were drawn to determine the specific effect of the factor on the drying coefficient.

### 3.6. Model

According to the results of the variance analysis and effect factor analysis, prediction models were established for the drying coefficients of the fuelbeds of white oak and Masson pine by selecting an appropriate model, with the drying coefficient as the dependent

variable and the significant effect factor (that is, the factor with a *p* value of less than 0.05 in the variance analysis) as the independent variable. The mean absolute error (MAE) and mean relative error (MRE) were calculated, and the calculation method is shown in Equations (5) and (6), where the two-stage error is the arithmetic mean of the two-stage model error.

$$MAE = \frac{1}{n}\sum_{i=1}^{n}|g_i - \hat{g}_i| \tag{5}$$

$$MRE = \frac{1}{n}\sum_{i=1}^{n}\frac{|g_i - \hat{g}_i|}{g_i} \tag{6}$$

where $g_i$ indicates the measured value of the drying coefficient ($h^{-1}$), $\hat{g}_i$ indicates the predicted value of the drying coefficient ($h^{-1}$), and n indicates the number of samples (n equals five in this study).

## 4. Results

### 4.1. Basic Information

Figure 1 shows the basic moisture content of white oak and Masson pine fuelbeds in all drying process experiments. The initial moisture content of the white oak fuelbeds varied from 98.67% to 104.58%, and the moisture content varied from 10.38% to 13.78% after the drying process experiment. For the Masson pine fuelbed, the minimum and maximum initial moisture contents in the drying experiment were 91.37% and 102.97%, respectively, and the variation range of moisture content after the experiment was 7.75–14.35%.

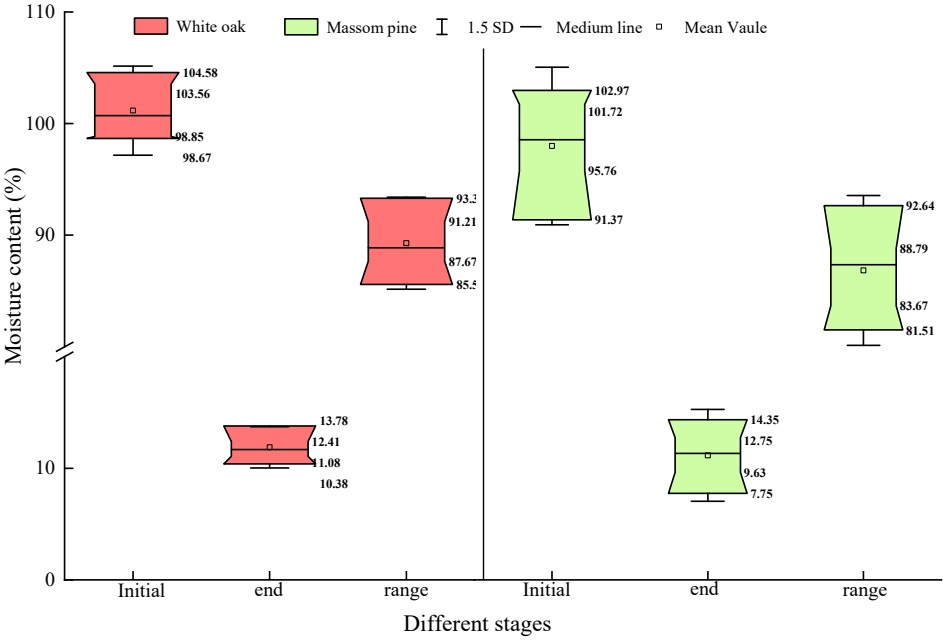

**Figure 1.** Range of moisture content of fuelbeds in the drying process experiment.

### 4.2. Drying Process

Under all the ratios of wind velocity and compactness, the drying process of white oak and Masson pine fuelbeds show an exponential downward trend. When the initial moisture content reaches 30%, the drying process can be regarded as a linear decline, but when the moisture content is lower than 30%, the drying process tends to gradually slow down, and the moisture content is basically the same at the end. The longest time from the initial moisture content to the fiber saturation point was approximately 1.8 h for the white oak fuelbed and 2.1 h for the Masson pine fuelbed (Figure 2).

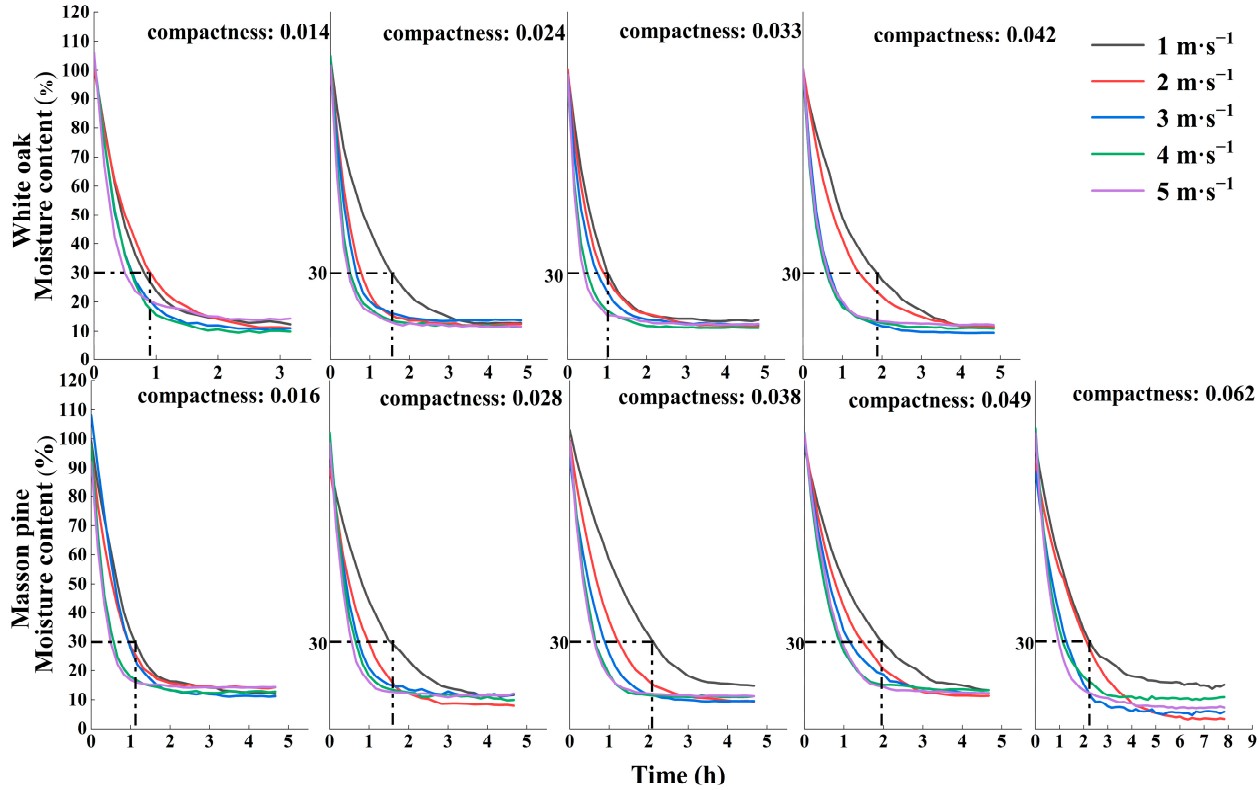

**Figure 2.** Drying process of white oak and Masson pine fuelbeds under different wind velocities and degrees of compactness.

### 4.3. Parameter Estimation and T-test

When the compactness range of the white oak is 0.014–0.042, the mean values of the drying coefficients $k$, $k_1$, and $k_2$ are 1.87, 1.88, and 1.63 h$^{-1}$, respectively. When distinguishing the drying process, the drying coefficient of the first stage is not significantly different from that of $k$ at all ratios, while the drying coefficient of the second stage is generally significantly lower than that of $k$. The drying coefficient of the Masson pine fuelbed is slightly lower than that of white oak, and the mean values of $k$, $k_1$, and $k_2$ are 1.50, 1.49, and 1.62 h$^{-1}$, respectively, when the compactness range of the Masson pine is 0.016–0.062. There is no significant difference between $k$ and $k_1$ at all ratios, but the mean value of $k_2$ is generally significantly higher than that of $k$ (Figure 3).

### 4.4. Effects of Wind Velocity and Compactness on Drying Coefficient

Table 3 shows the effects of wind velocity and bed compactness on the drying coefficient. The wind velocity, compactness, and their interaction have extremely significant effects on the three drying coefficients in both the white oak and Masson pine fuelbeds.

**Table 3.** The ANOVA results.

| Fuel Type | Index | df | $k$ | | $k_1$ | | $k_2$ | |
|---|---|---|---|---|---|---|---|---|
| | | | **F Value** | *p* | **F Value** | *p* | **F Value** | *p* |
| | Wind | 4 | 355.000 | *** | 370.359 | *** | 27.568 | *** |
| White oak | Compactness | 3 | 41.751 | *** | 43.459 | *** | 12.590 | *** |
| | Wind × compactness | 12 | 13.094 | *** | 13.285 | *** | 7.843 | *** |
| | Wind | 4 | 242.774 | *** | 277.675 | *** | 32.269 | *** |
| Masson pine | Compactness | 4 | 110.449 | *** | 126.444 | *** | 27.111 | *** |
| | Wind × compactness | 16 | 4.760 | *** | 6.071 | *** | 2.638 | ** |

**Note:** *** indicates $p < 0.001$; ** indicates $p < 0.01$.

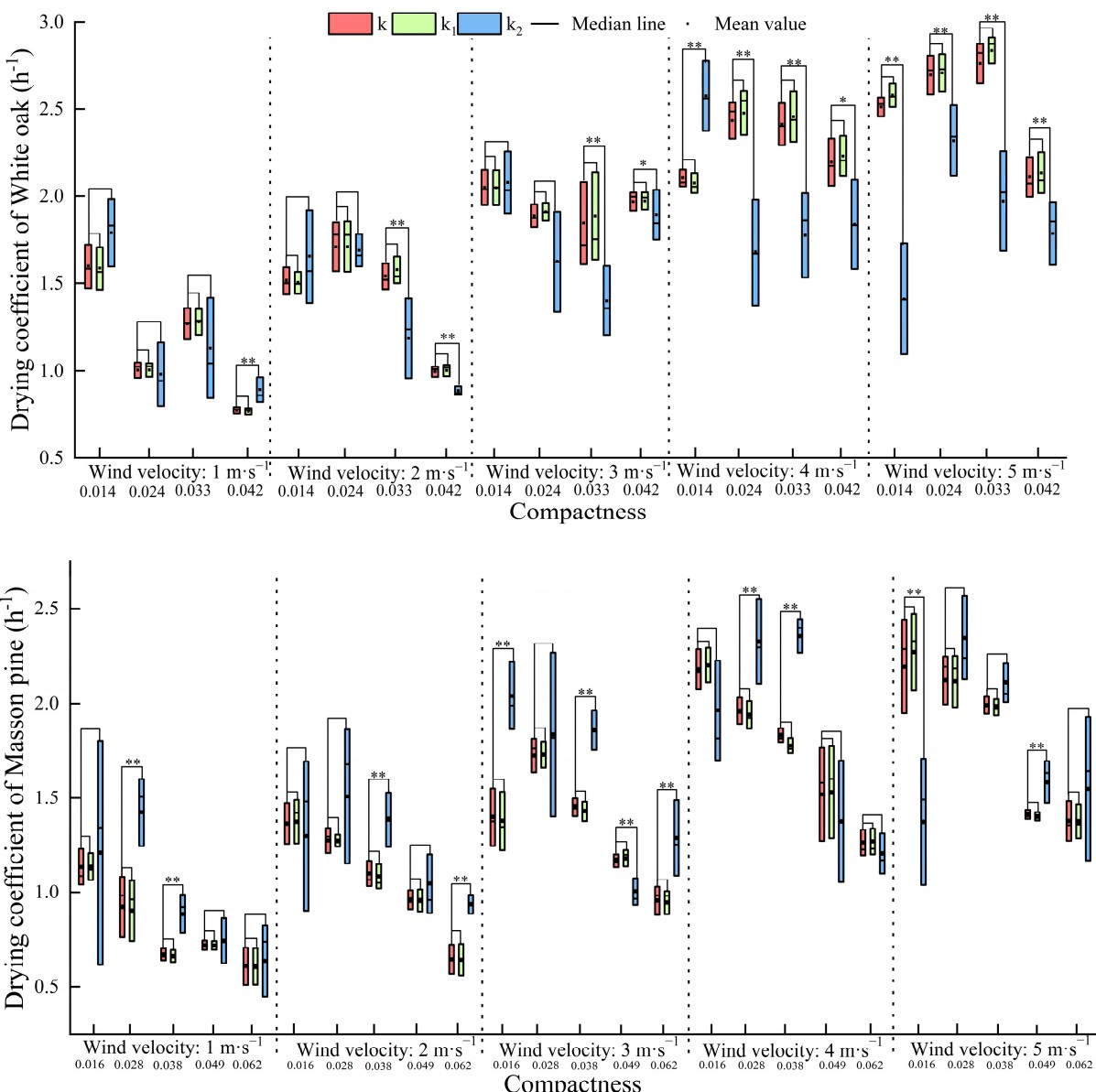

**Figure 3.** *T*-test results for different drying coefficients for white oak and pine fuelbeds. Note: ** indicates there is a very significant difference between the two drying coefficient; * indicates there is a significant difference between the two drying coefficient.

Figure 4 shows the variation in the drying coefficient of the white oak fuelbeds at different degrees of compactness and wind velocities. Regardless of how the compactness and wind velocity change, $k$ and $k_1$ follow the same trend. When the wind velocity is lower than $3 \text{ m·s}^{-1}$, the three drying coefficients decrease with increasing compactness. When the wind velocity is $3 \text{ m·s}^{-1}$, the drying coefficient between different degrees of compactness has no significant difference. When the wind velocity is more than $3 \text{ m·s}^{-1}$, $k_2$ and the other two drying coefficients show different trends. When the compactness is fixed, the three drying coefficients all show an increasing trend with increasing wind velocity, in which $k$ and $k_1$ show the same trend with increasing wind velocity. The effect of wind velocity on $k_2$ is weak, and the difference in $k_2$ between adjacent wind velocities is not significant.

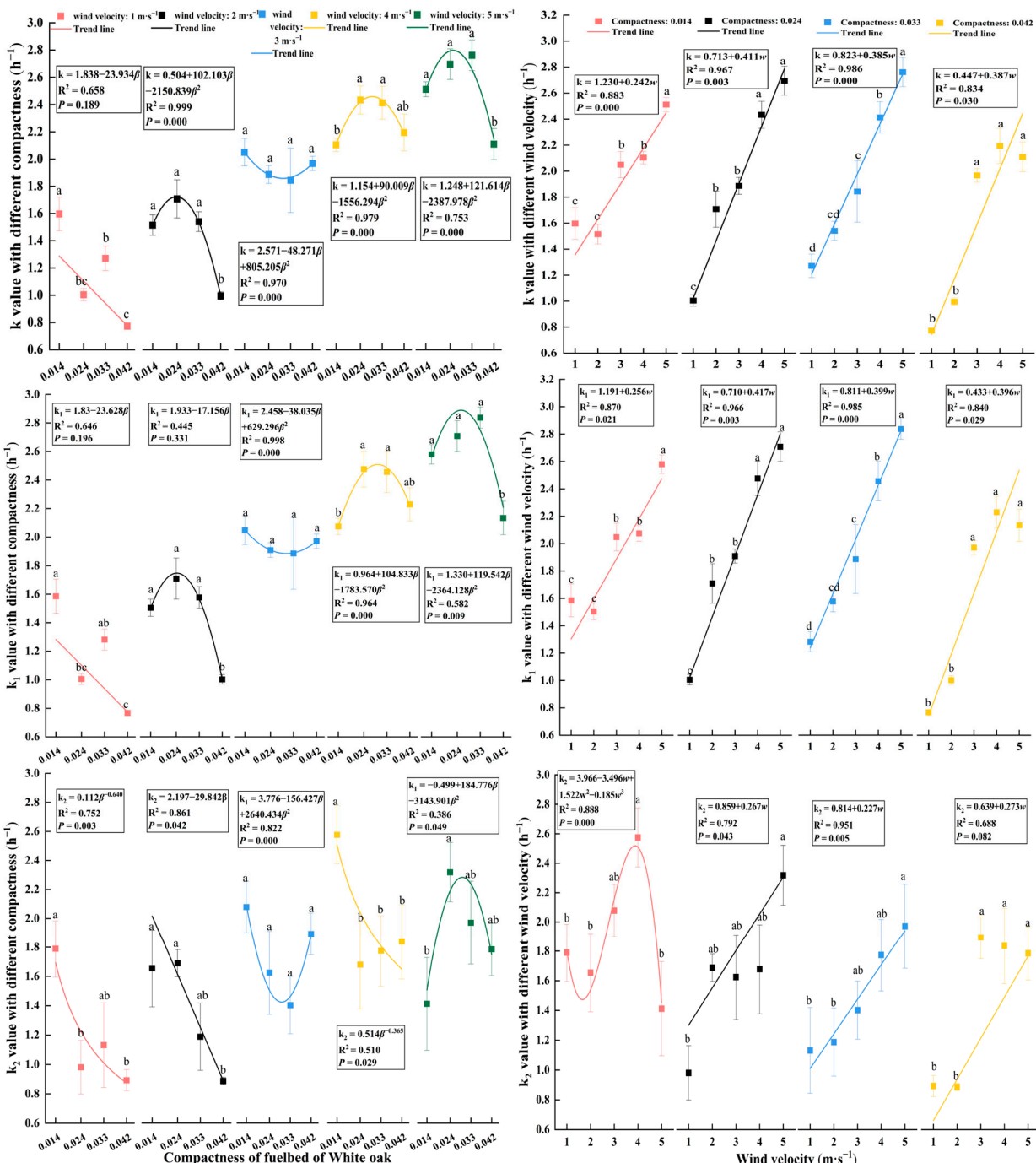

**Figure 4.** Variation in the drying coefficient of the fuelbed of white oak at different degrees of compactness and wind velocities. Note: $k$, $k_1$, $k_2$ indicates the drying coefficient, $w$ indicates the wind velocity, and $\beta$ indicates the compactness. The same below.

The variation trend of $k$ and $k_1$ in Masson pine fuelbeds is the same regardless of the change in wind velocity and compactness. When the wind velocity is fixed, the three dying coefficients all show a downward trend with increasing compactness of the fuelbed, but the difference in $k_2$ among different compactness values is not as obvious for the other two dying coefficients. When the compactness is fixed, the fuelbed drying coefficient increases with increasing wind velocity. When the wind velocity is 3 m·s$^{-1}$, the increasing trend of the three drying coefficients in the father (the highest slope) and the effect of wind velocity on $k_2$ are weaker than those of the other two drying coefficients (Figure 5).

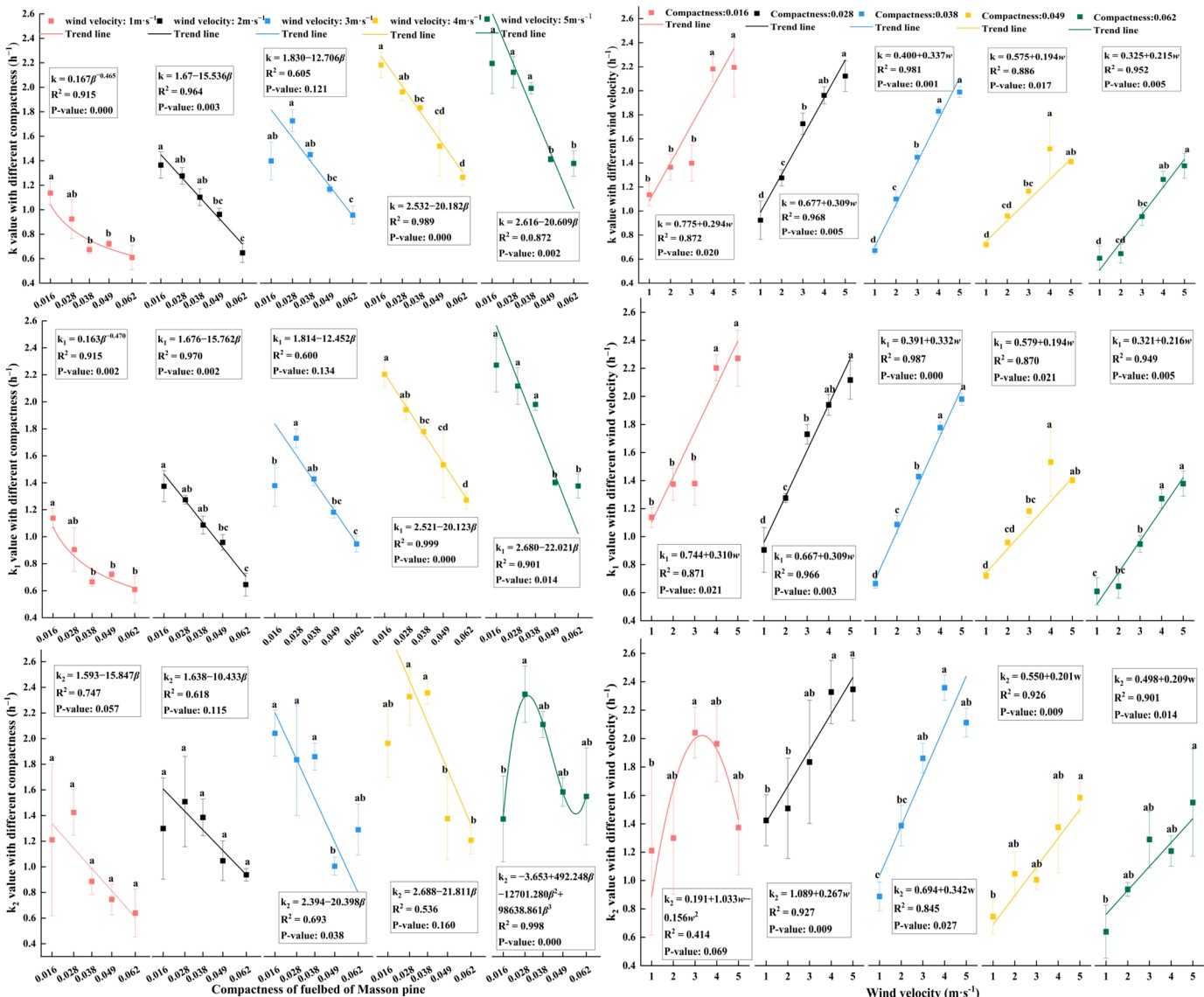

**Figure 5.** Variation in the drying coefficient of the fuelbed of pine at different degrees of compactness and wind velocities.

*4.5. Model*

4.5.1. Parameters of the Model

Taking compactness as the classification condition, the prediction models of the drying coefficient of white oak and Masson pine fuelbeds with wind velocity were established (Table 4). The minimum and maximum relative errors of *k* of white oak are 6.20% and 17.46%, respectively. The mean relative errors of $k_1$ and $k_2$ range from 3.45% to 14.13% and 2.72% to 15.36%, respectively. The mean relative error ranges of *k*, $k_1$, and $k_2$ for Masson pine fuelbeds are 4.50–7.48%, 4.27–8.08%, and 4.18–7.32%, respectively. In both the broadleaf and needleleaf beds, the fitting effect of the drying coefficient prediction model with different drying stages is slightly better than that without different drying stages.

**Table 4.** Results of the prediction models.

| Fuel Type | Compactness | $k$ Model | $R^2$ | Mae (h$^{-1}$) | Mre (%) | $k_1$ Model | $R^2$ | Mae (h$^{-1}$) | Mre (%) | $k_2$ Model | $R^2$ | Mae (h$^{-1}$) | Mre (%) |
|---|---|---|---|---|---|---|---|---|---|---|---|---|---|
| White oak | 0.014 | $k = 1.230 + 0.242\,w$ | 0.883 | 0.117 | 6.58 | $k_1 = 1.191 + 0.256\,w$ | 0.870 | 0.135 | 7.46 | $k_2 = 0.185w^3 + 1.522w^2 - 3.496w + 3.966$ | 0.888 | 0.055 | 2.72 |
|  | 0.024 | $k = 0.713 + 0.411\,w$ | 0.967 | 0.100 | 6.20 | $k_1 = 0.710 + 0.417\,w$ | 0.966 | 0.104 | 6.33 | $k_2 = 0.859 + 0.267\,w$ | 0.792 | 0.170 | 10.93 |
|  | 0.033 | $k = 0.823 + 0.385\,w$ | 0.986 | 0.403 | 17.46 | $k_1 = 0.811 + 0.399\,w$ | 0.985 | 0.061 | 3.45 | $k_2 = 0.814 + 0.227\,w$ | 0.951 | 0.068 | 5.08 |
|  | 0.042 | $k = 0.447 + 0.387\,w$ | 0.834 | 0.224 | 14.16 | $k_1 = 0.433 + 0.396\,w$ | 0.840 | 0.225 | 14.13 | $k_2 = 0.639 + 0.273\,w$ | 0.688 | 0.205 | 14.36 |
| MRE |  | 11.10 |  |  |  | 7.84 |  |  |  | 8.27 |  |  |  |
| Masson pine | 0.016 | $k = 0.775 + 0.294\,w$ | 0.857 | 0.122 | 7.47 | $k_1 = 0.744 + 0.310\,w$ | 0.916 | 0.126 | 8.08 | $k_2 = 0.191 + 1.033\,w - 0.156w^2$ | 0.914 | 0.076 | 4.18 |
|  | 0.028 | $k = 0.677 + 0.309\,w$ | 0.957 | 0.070 | 4.50 | $k_1 = 0.667 + 0.309\,w$ | 0.633 | 0.070 | 4.50 | $k_2 = 1.089 + 0.267\,w$ | 0.947 | 0.097 | 5.20 |
|  | 0.038 | $k = 0.400 + 0.337\,w$ | 0.980 | 0.062 | 4.77 | $k_1 = 0.391 + 0.332\,w$ | 0.354 | 0.052 | 4.27 | $k_2 = 0.694 + 0.342\,w$ | 0.834 | 0.093 | 4.84 |
|  | 0.049 | $k = 0.575 + 0.194\,w$ | 0.972 | 0.072 | 5.61 | $k_1 = 0.579 + 0.194\,w$ | 0.574 | 0.081 | 6.36 | $k_2 = 0.550 + 0.201\,w$ | 0.887 | 0.060 | 5.59 |
|  | 0.062 | $k = 0.325 + 0.215\,w$ | 0.925 | 0.059 | 7.48 | $k_1 = 0.321 + 0.216\,w$ | 0.292 | 0.063 | 7.89 | $k_2 = 0.498 + 0.209\,w$ | 0.866 | 0.078 | 7.32 |
| MRE |  | 5.97 |  |  |  | 6.22 |  |  |  | 5.43 |  |  |  |

Note: The mean absolute error and relative error of the prediction models for the white oak fuelbed drying coefficients that did not distinguish and that did distinguish the drying process are 0.211 h$^{-1}$ and 11.1% and 0.129 h$^{-1}$ and 8.18%, respectively. The mean absolute error and relative error of the prediction models for Masson pine fuelbed the drying coefficients that did not distinguish and that did distinguish the drying process are 0.077 h$^{-1}$ and 5.967% and 0.080 h$^{-1}$ and 5.82%, respectively.

### 4.5.2. A 1:1 Comparison

Figure 6 shows 1:1 figures of the measured and predicted values of the three prediction models for the drying coefficients of white oak and pine fuelbeds. The prediction error of the $k$ value for white oak is largely underestimated when the compactness is 0.033, and the measured and predicted values can be evenly distributed on both sides of the 1:1 line in other cases, with a good prediction effect. The prediction results for $k$ and $k_1$ for the Masson pine fuelbeds are good, while the prediction results for $k_2$ are mainly underestimated.

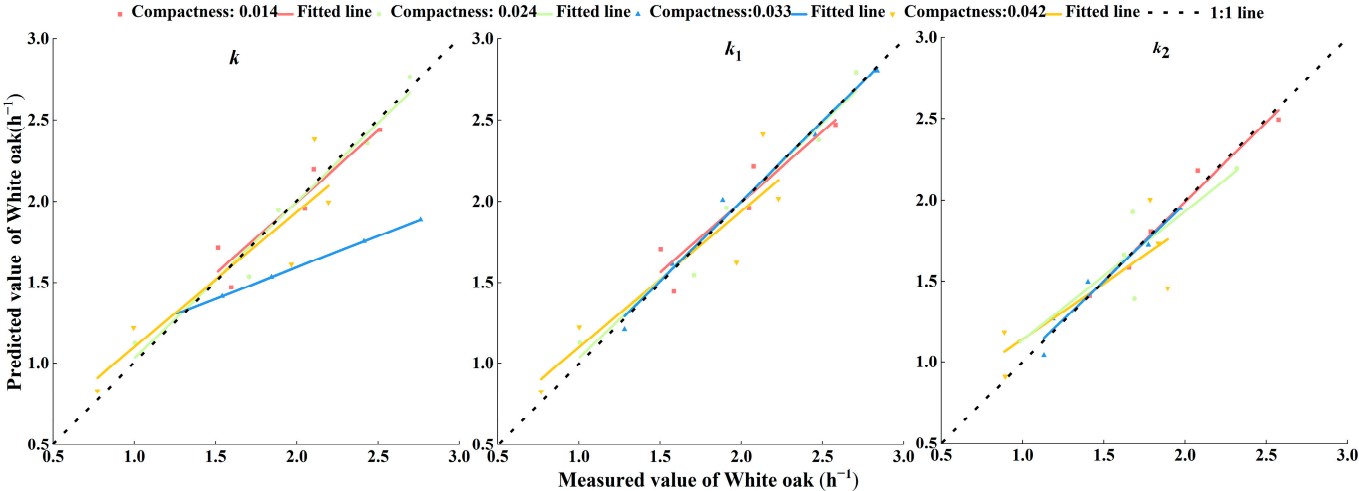

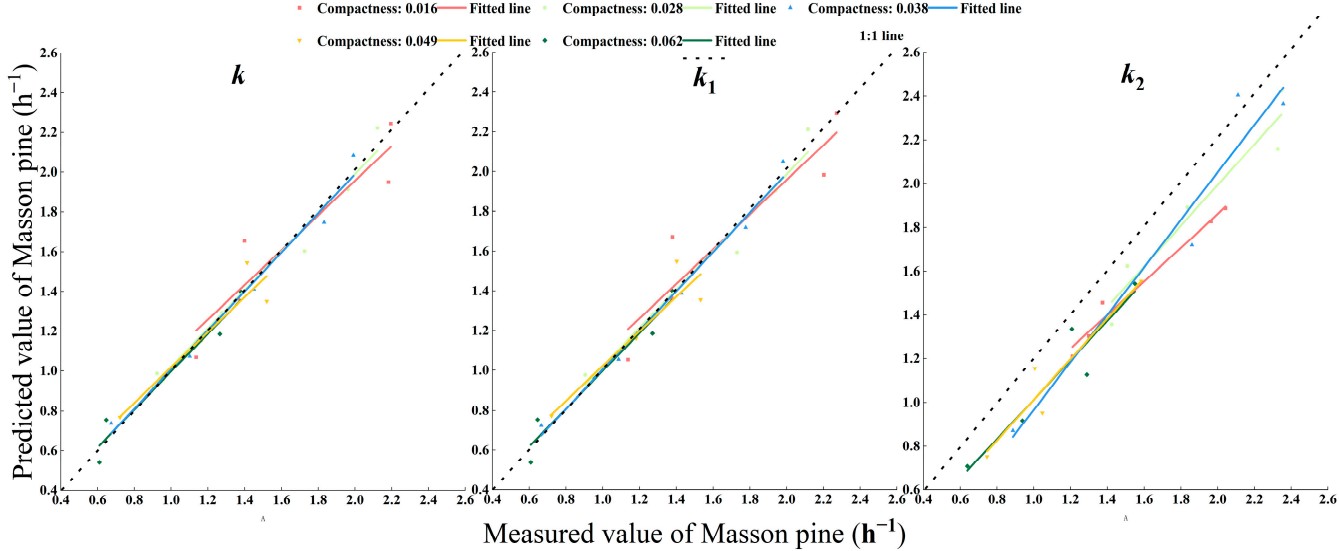

**Figure 6.** 1:1 diagram of measured and predicted values.

## 5. Discussion

### 5.1. Basic Information concerning the Drying Coefficient

Regardless of how wind velocity and compactness changed, under the conditions of approximately constant temperature and humidity, the moisture content of the fuelbed showed an exponential downward trend and, eventually, tended to be consistent; these results are similar to those of Zhang [12]. If the drying process was not distinguished, the drying coefficients of the white oak and Masson pine fuelbeds were $1.869 \pm 0.073$ h$^{-1}$ and $1.356 \pm 0.057$ h$^{-1}$, respectively. Zhang et al. found that the drying coefficient for Mongolian oak fuelbeds was $0.731 \pm 0.237$ h$^{-1}$ [32]. Jin et al. tested red pine needles

and found that the drying coefficient was $0.605 \pm 0.243\,\text{h}^{-1}$ [14]. Both were lower than those in this study. This is mainly due to the different degree of compactness of the fuelbed. In this study, the compactness ranges of white oak and pine were 0.014–0.042 and 0.016–0.062, respectively, while the compactness ranges of Mongolian oak and red pine in Jin's study were only 0.0092–0.0184 and 0.0158–0.0316, respectively, both significantly lower than those in this study. Zhang found that the drying coefficient of broadleaf beds was significantly higher than that of needleleaf beds [33]. In this study, the same conclusion was found: the drying coefficient of white oak was significantly higher than that of Masson pine ($t = 5.596$, $p = 0.000$), mainly due to the different structures and physicochemical properties of needle-bearing and broadleaf trees. In addition, according to the fitting curve of the drying coefficient in Figures 4 and 5, it can be seen that the drying coefficient mechanism for white oak and Masson pine fuelbeds was also different, further indicating that employing the same drying coefficient model for broadleaf and needleleaf litter may cause significant errors.

The $k_1$ and $k_2$ of the white oak fuelbeds were $1.887 \pm 0.075\,\text{h}^{-1}$ and $1.628 \pm 0.062\,\text{h}^{-1}$, respectively. The $k_1$ and $k_2$ of Masson pine fuelbeds were $1.357 \pm 0.057\,\text{h}^{-1}$ and $1.493 \pm 0.062\,\text{h}^{-1}$, respectively. The $k_1$ of white oaks was significantly higher than that of Masson pine ($t = 5.723$, $p = 0.000$), while $k_2$ was not significantly different between the two litter types ($t = 1.523$, $p = 0.130$). However, the variation in the drying coefficient of the two litter types with wind velocity and compactness was different, indicating that the drying mechanisms of the two litter types are different. Even if the drying process is distinguished, the same drying coefficient prediction model should not be used for both broadleaf and needleleaf crops. In particular, the current widely used drying coefficient models are easily obtained, using pine needles or humidity rods with homogeneous structures as research objects, and failure to distinguish litter types is one of the main sources of error in predicting litter moisture content.

*5.2. Difference Analysis*

In both white oak and Masson pine fuelbeds, there was no significant difference between $k$ and $k_1$, and there was a significant difference between $k$ and $k_2$, under different degrees of compactness and wind speed ratios. Although there was no significant difference between $k$ and $k_1$, the maximum difference was 4.50% for white oak and 7.05% for Masson pine. The $k_2$ of white oak fuelbeds was significantly lower than that of $k$, and the maximum variation was 72.76%. The significant difference between $k$ and $k_2$ was mainly due to the different drying mechanisms of evaporation and diffusion. According to the studies of Pippen and Zhang, under conditions of fixed temperature and humidity and no wind, a downward trend in free water is easier to achieve than in bound water, and evaporation is faster than diffusion under the same conditions [12,34]. Therefore, in theory, $k$ or $k_1$ should be higher than $k_2$. The $k_2$ of pine fuelbeds was indeed higher than $k$, which may be because needles have a large surface-area-to-volume ratio and increasing wind velocity has a greater effect on needles. Therefore, the decrease in needle temperature hinders evaporation and significantly promotes diffusion. Therefore, $k_2$ will be higher than $k$.

The $k$ and $k_2$ of the two litter types were significantly different, indicating that the undistinguishing drying process was one of the main sources of error in the prediction of litter moisture content, especially when the litter moisture content was less than 30%, which may cause significant error and has a serious impact on the accuracy of forest fire risk and fire behavior prediction.

*5.3. Impact Factor Analysis*

Wind velocity, compactness, and their interaction had significant effects on the $k$, $k_1$, and $k_2$ of the two litter beds. As shown in Figures 4 and 5, with increasing wind velocity, $k$ and $k_1$ of the two litter beds showed an overall increasing trend but did not increase significantly at $4\,\text{m}\cdot\text{s}^{-1}$ and $5\,\text{m}\cdot\text{s}^{-1}$. Even when the bed compactness of white oak fuelbeds was 0.042 and that of pine fuelbeds was 0.049, there was a downward trend. This was

mainly due to the duality of wind velocity's effect on the evaporation process [18]. On the one hand, it will remove water vapor from litter surfaces and reach the litter's internal water, but when the wind velocity is too high, the water on the surface of the litter will spread too fast, causing surface hardening of the litter and reducing the drying rate [35]. Therefore, it can be predicted that when the bed compactness is constant, the continuous increase in wind velocity will reduce the drying coefficient. When the compactness is very low (that of white oak and Masson pine is 0.014 and 0.016), the impact of low wind velocity is weak; thus, the drying process curves of different wind velocities will cross or overlap, without obvious regularity, and even the drying rate of 2 m·s$^{-1}$ will be lower than 1 m·s$^{-1}$.

With increases in bed compactness, the $k$ and $k_1$ of the two litter beds showed a general downward trend; this was mainly because with increasing bed compactness, the more compact the litter monomer in the bed, the more complex the outward diffusion path of water in the bed, thus reducing the drying coefficient [36,37](Groot and Wardati, 2005; Mahapatra 2011). The effect mechanisms of bed compactness in the two litters on the evaporation process was different (different fitting curves in Figures 4 and 5), which may have been caused by the large difference in bed structure due to the different shapes of the coniferous and broadleaf litter.

The variation trend of $k_2$ with wind velocity and compactness was different from that of $k$ and $k_1$, which further indicates that the mechanisms of the two drying processes were different. It is necessary to distinguish the drying processes when using semiphysical methods to predict litter moisture content. Although the variation trend of $k_2$ with varying wind velocity and compactness in the two fuelbeds was basically the same as that of $k$ and $k_1$, there was no significant difference between the adjacent gradients, and the variation range was lower than that of $k$ and $k_1$. This may be because the second part of the drying process involves the thermal movement of water molecules, which is related to temperature [6,38,39], and according to this experiment, the temperatures of the litters did not change significantly; so, the range of the $k_2$ change was small.

*5.4. Prediction Models*

Prediction models of the drying coefficient of white oak and Masson pine fuelbeds based on bed compactness were established, with the MREs of all models being lower than 15%, all within the allowable error range [12]. The MREs of white oak and pine were 11.10% and 5.97%, respectively, while the average MREs of the two stages were 8.06% and 5.83%, respectively, when the drying process was distinguished; this indicates that the process of drying could better fit the drying equation. The prediction effect of pine needles was significantly better than that of white oak broad leaves, mainly due to the greater homogeneity of needles compared with broad leaves [40] and the small variability in moisture content change.

## 6. Conclusions

The simplification of the drying process and the disregard for fuelbed structure may be among the main sources of error in the prediction of litter moisture content based on the semiphysical method. To further improve the accuracy of the prediction model, the drying processes of white oak and Masson pine fuelbeds with different degrees of compactness were measured at different wind velocities, and the fiber saturated moisture content (30%) was taken as the cutoff point. The differences in the drying coefficient and effect factors were analyzed when the drying process was not distinguished and when it was distinguished. The results show that there were significant differences in the drying coefficients of fuelbeds with different wind velocities and compactness, and the drying processes and mechanisms of broadleaf and needleleaf litters were also different. Simplifying the drying process and not considering the fuelbed structure and litter type lead to a certain error in the moisture content. The results from the drying coefficient prediction model based on the distinguishing drying process are better than those of the model without a distinguishing drying process, which further indicates that the distinguishing drying



process can improve the accuracy of predictions. This study employs indoor simulation research to measure the specific impact of wind velocity and fuelbed structure on the drying coefficient of broadleaf and needle beds when distinguishing the drying process and establishing prediction models, respectively, which is of theoretical significance and lays a foundation for the later improvement of moisture content accuracy based on the semiphysical method; however, these results have not been verified in the field. In future research, different models and methods (such as the Arrhenius model, which is more practical [41]) should be selected for field experiments and correction experiments. In addition, in this study, the drying process was artificially divided into two stages by taking 30% as the cutoff point according to the drying curve, which may also have caused some errors. In future studies, it will be necessary to accurately determine the moisture content of the drying mechanism transformation under different wind velocity conditions; this would be of great significance for improving the accuracy of moisture content and forest fire risk prediction.

**Funding:** This research was funded by Guizhou Provincial Science and Technology Projects grant number (ZK [2021] general 158), the China National Natural Science Foundation grant number (32201563), and the Guizhou Provincial Science and Technology Projects grant number (Qianke Support [2022] General 249).

**Institutional Review Board Statement:** Not applicable.

**Informed Consent Statement:** Not applicable.

**Data Availability Statement:** Not applicable.

**Conflicts of Interest:** The author declares no conflict of interest.

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
