# Peer review of "Indoor Experiments on the Moisture Dynamic Response to Wind Velocity for Fuelbeds with Different Degrees of Compactness"

_fire, doi:10.3390/fire6030090_

Round 1
Reviewer 1 Report
Please see the attached file.

Author Response
The author conducted a series of indoor experiments to establish a model for predicting drying dynamics of fuelbeds of white oak and masson pine with different levels of compactness when exposed to wind.
In the process of creating the fuelbeds, the author considered the actual conditions by sampling fuelbeds from the outdoor study are in Tianhe Mountain. After creating the fuelbeds with different compactness levels, the fuelbeds were exposed to 5 different wind velocities and recording of temperature and relative humidity were made.
In the proposed model by the author, there were 3 drying coefficients, k, k1 and k2 representative of 3 drying stages. Results for these 3 coefficients as a function of type of fuelbed, compactness and wind velocity were presented. Statistical analysis were also preformed on the data obtained from the experiments. The presented results were meant to explain the drying dynamics and its dependence on the fuelbed type, compactness and wind velocity.
Overall, this paper can contribute positively to the knowledge in this field. However, the author needs to address the below comments in order to improve the quality of paper so that it can be published.
Answer:Thank you for your revision. All revisions have been made as required.
- Please revise the paper title. Something like the following would be more clear: “Indoor experiments on the moisture dynamic response to wind velocity for fuelbeds with different degrees of compactness”
Answer: Thank you for your advice. The title of the paper has been revised as required.
- Please improve the English language quality of the manuscript. It needs a significant improvement. Some examples are:
- Line 12: “… and non-consideration of the fuelbed structure” should change to “… and not accounting for the fuelbed structure”
- Line 15: “factors influencing” should change to “influencing factors”
- Line 132: Section 2.2 title should change to “Constructing of indoor fuelbeds with different compactness levels.”
- Line 165: “five gradients of wind velocity” should change to “five wind velocity values of”
- Timelag should change to Time-lag
Answer: Thank you very much. It has been revised according to your comments and has been polished by a professional language polishing agency.
- Line 15: “takes the fiber saturation point as the boundary”. Please briefly elaborate on the boundary here. It is boundary between what and what?
Answer: Thank you for your advice. In this manuscript, the boundary is the saturated fiber moisture content (30%), that is, when the moisture content is 30%, the drying process is divided into two stages: from initial moisture content to the saturated moisture content and from the saturated moisture content to the equilibrium moisture content.
It has been added to the manuscript.
- Table 1: What is DBH? Please clarify either in the text of in the table.
Answer: DBH indicates the diameter at the breast height of the tree, which has been modified in the manuscript.
- Section 2.1: can the author provide information about when (what time of the year) they took samples of fuel bed from the study area?
Answer: Sorry to ignore this important information, thank you for your comment. The time of field sample collection is February 2022, which has been added to the manuscript.
- Line 147-155: the author does a good job in providing a definition for the compactness. I would recommend putting this definition in Section 2.1 where the author presents compactness data in Table 1.
Answer: Thank you for your advice. This content has been put in Section 2.1.
- Line 148: Please replace Formula (1) with Equation (1). Do this wherever applicable.
Answer: Thank you for your advice. The Formula in the manuscript has been replaced with Equation.
- Line 149: The author mentions some references relevant to the values of particle density. Please cite those reference within the text.
Answer: Thank you for your advice. Reference has been added as required.
- Line 182: In Formula (2), what does the Greek letter tau (in the denominator) represent? Where is k in this formula?
Answer: Thank you, there’s an error here, it should be the Greek letter tau, which indicates the Time-lag, and has been modified in the manuscript.
- The moisture content that the author is referring to is on wet-mass or dry-mass basis?
Answer: Thank you for your advice. The moisture content is on a dry-mass basis, which has been added to the manuscript.
- In equations (6) and (7), what is the value for n in the summation term?
Answer: n indicates the number of samples. In this study, n is same as the number of wind velocity gradients, which is 5.
- The exponential behavior of the moisture content in Figure 2 is an interesting observation. I would recommend more elaboration on this behavior by paying attention to the following items:
(I) Equation (4): By assuming that E is not a function of time, and integrating, an exponential relation for M can be obtained.
(II) A mathematical model to express moisture evaporation is called Arrhenius model. When this model is used, it results in smooth moisture behavior inside the fuel particle [1]. The connection between this and the results of Figure 2 is interesting and worth mentioning in the paper.
[1] Borujerdi PR, Shotorban B, Mahalingam S, Weise DR. Modeling of water evaporation from a shrinking moist biomass slab subject to heating: Arrhenius approach versus equilibrium approach q. International Journal of Heat and Mass Transfer. 2019;145:118672.
Answer: Thank you very much for your suggestion. I have carefully read the reference that you provided and have benefited a lot. The main purpose of this study is to verify that simplifying the drying process and ignoring the fuelbed structures are the reasons for the error in moisture content prediction, so it is necessary to select a relatively stable variable for analysis. As mentioned in the manuscript, the drying coefficient calculated in this study is only related to the fuel type and the bed structure, which can be better analyzed.
You proposed two methods to fit the drying process under different wind velocity and bed compactness (Fig.2). At this time, the drying rate is obtained, unlike the drying coefficient, it is also affected by the initial moisture content, the moisture content and equilibrium moisture content at the previous moment. Therefore, if this method is selected in this study, the next study can not be carried out. For example, the drying rate can be calculated by distinguishing the two drying processes, but because the initial moisture content of the two dying process is different, it is bound to affect the two dying rates and cannot be compared. The two methods you proposed may better simulate the drying process, but the next analysis cannot be carried out, so no relevant research has been carried out.
However, the two methods provide new idea for the prediction of moisture content in the field. In the follow-up research, these methods will be considered to calculate the drying rate for moisture content prediction and compare the accuracy. The discussion of relevant contents is also added in the manuscript. Thank you again for your comments.
- Figure 2: The scale on the x-axis is 0.5. This causes the numbers to get very close to each other. I would recommend changing the scale to 1 so that the figure can be read more easily.
Answer: Thank you for your advice. Figure 2 has been modified as required.
- Figure 2: For white oak with compactness of 0.014, why is drying faster for case with 1 m/s compared with case with 2 m/s? For all other cases, drying is faster for higher velocity and this makes sense.
Answer: The study shows that the influence of wind velocity on the drying coefficient is affected by the fuelbed compactness. When the compactness is very low (white oak and Masson pine density of 0.014 and 0.016), the influence of low wind velocity is weak, so the drying process curves of different wind velocities will cross or overlap, without obvious regularity, so this situation will occur. Also added a discussion in the manuscript.
- Figure 4: Please change light green color of one of the plots to another color (black, for example). Also for the k values vs compactness graphs, the numbering of x-axis is really small and hard to read. Please address this issue.
Answer: Thank you for your advice, the light green has been replaced with black, and the x-axis enlarged to read.
- Same issues as of Figure 4 for Figure 5 too. Please note that, the figures are very important part of your paper. Therefore, please do your best in presenting high quality figures.
Answer: Thank you for your advice, the light green has been replaced with black, and the x-axis enlarged to read.

Reviewer 2 Report
The objective of the paper is to measure the characteristic drying times of two littler fuel beds.
The paper is not understandable in the present form. The authors spend a significant space on simple concepts but glaze over important more complicated ones. References for important terms are not provided.
Some specific comments, certainly not a complete list:
Line 12: Better: a lack of consideration
Line 49: None of the three references report error within 1% point as claimed here. Also, Carlson et al. (2007) report much bigger errors. Please reconcile. Clarify which kind of error (e.g., bias or variance).
Line 152 and all other equations: There should be no indent after an equation, and the equation numbers should be on the right not immediately after the formula
Line 164: Should be all bold
Line 165: “five gradients of wind velocity were set” -> “five values of wind velocity were considered”
Line 168: “set” -> “selected”
Line 184: There is no k in (2)
Line 185: Do not start a sentence with a formula or a symbol
Line 185 and the rest of the page: why only drying, what if the initial M < E ?
Line 190: “the conclusion is not general” reformulate
Line 187-193: do we really need this long explanation of the solution of the textbook ODE (2)?
Line 206: the same equation 3rd time, reformulate the page
Line 209: units missing after 2.3
Line 214: cite the other 3 out of the 4 calculation methods
Line 215: cite a reference to justify the “high accuracy” and quote number what that is
Line 234: what nonparametric estimation? It seems k is computed from 2 times on the exponential curve. This is of course true only if E is constant between the two times. Why should that be so? T and H can change quickly. Do you have measurements to show that T and H did not vary during the experiment?
Line 250: distinguished: English. Maybe something like defined piecewise.
Line 262: what is “significant influence factor” ?
Lines 265 and 266: conflict of notation with the k_1 and k_2 above. Choose a different notation. Are (6) and (6) also applied to the “distinguished” drying coefficient values?
Line 263: “n-fold method” is not how one would normally call in English a computation of the mean error from a sample of size n.
Line 294: the drying coefficients are not of much interest without knowing more about the fuel beds, such as the distribution of the particle sizes and shape, and the packing ratio. That should be defined and specified here.
Line 305: Define fuel bed compactness and provide a reference. Packing ratio is normally used to measure compactness, which is a qualitative term, not quantitative.
Line 306: Define how is the “influence” computed and provide a reference. Maybe some kind of influence coefficients? “influence” is used elsewhere in the paper in the common English sense. Confusing.
Line 353: Numbering of the lines ends. Please number the lines all the way to the end.
Section 5.1, comparing the drying coefficients: the results are about twice less than the cited one. This is not “slightly”.
Author Response
The objective of the paper is to measure the characteristic drying times of two littler fuel beds.
The paper is not understandable in the present form. The authors spend a significant space on simple concepts but glaze over important more complicated ones. References for important terms are not provided.
Some specific comments, certainly not a complete list:
Answer: Thank you for your comments. Which have been carefully revised according to your requirements.
- Line 12: Better: a lack of consideration
Answer: Thank you for your advice. The description has been modified:“Compared with the model without distinguishing the drying process, the MRE of the prediction models for white oak and Masson pine decreased by 27.39% and 2.35%, respectively. The prediction accuracy of the model of the drying coefficient obtained by distinguishing the drying loss process is higher than that of the model without distinguishing the drying process.”
- Line 49: None of the three references report error within 1% point as claimed here. Also, Carlson et al. (2007) report much bigger errors. Please reconcile. Clarify which kind of error (e.g., bias or variance).
Answer: Thank you. It has been revised as required, and what kind of error is clarified. Carlson’s references have been deleted.
- Line 152 and all other equations: There should be no indent after an equation, and the equation numbers should be on the right not immediately after the formula
Answer: Thank you for your advice. It has been revised as required.
- Line 164: Should be all bold
Answer: Thank you for your advice. It has been revised as required.
- Line 165: “five gradients of wind velocity were set” -> “five values of wind velocity were considered”
Answer: Thank you for your advice. It has been revised as required.
- Line 168: “set” -> “selected”
Answer: Thank you for your advice. It has been revised as required.
- Line 184: There is no k in (2)
Answer: Thank you, there’s an error here, it should be the Greek letter tau, which indicates the Time-lag, and has been modified in the manuscript.
- Line 185: Do not start a sentence with a formula or a symbol
Answer: Thank you for your advice. It has been revised as required.
- Line 185 and the rest of the page: why only drying, what if the initial M < E ?
Answer: The main purpose of this study is to analyze how wind velocity affects the moisture change of fuel when distinguishing the moisture process and considering the fuelbed structure. At this time, it is a drying process, and forest fire managers also pay more attention to the drying process of fuel, so this study mainly conducts the drying study. When the moisture content of fuel drops to equilibrium moisture content, the experiment is stopped, so there is no initial M < E.
Thank you for your question, in the follow-up study, the change of moisture content of fuel in different bed structures under different temperatures and humidity will be simulated indoors, when initial M < E occurs, the moisture content will increase.
- Line 190: “the conclusion is not general”reformulate
Answer: Thank you for your advice. It has been revised as required.
- Line 187-193: do we really need this long explanation of the solution of the textbook ODE (2)?
Answer: Thank you for your advice. It is a litter wordy and has been rewritten.
- Line 206: the same equation 3rdtime, reformulate the page
Answer: Thank you for your advice, It has been revised as required.
- Line 209: units missing after 2.3
Answer: Thank you, It has been added.
14.Line 214: cite the other 3 out of the 4 calculation methods
Answer: Thank you, It has been added.
- Line 215: cite a reference to justify the “high accuracy” and quote number what that is
Answer: Thank you for your advice. It has been added.
- Line 234: what nonparametric estimation? It seems k is computed from 2 times on the exponential curve. This is of course true only if E is constant between the two times. Why should that be so? T and H can change quickly. Do you have measurements to show that T and H did not vary during the experiment?
Answer: Thank you, the first problem is my writing error. It should be nonlinear parameter estimation instead of nonparametric estimation, which has been modified in the manuscript.
The T and H are recorded synchronously each time the mass of the fuelbed is weighted, which has been described in the manuscript: “The mean temperature variation of the experiments in this study was 2.3℃, and the mean relative humidity variation was 1.6%, which could be considered approximately constant temperature and humidity (Jin et al., 2016).”
- Line 250: distinguished: English. Maybe something like defined piecewise.
Answer: Thank you for your advice. It has been revised.
- Line 262: what is “significant influence factor” ?
Answer: The significant influence factor indicates the factor with p value less than 0.05 in the analysis of variance, which has been added in the manuscript.
- Lines 265 and 266: conflict of notation with the k_1 and k_2 above. Choose a different notation. Are (6) and (6) also applied to the “distinguished” drying coefficient values?
Answer: Thank you for your advice, It has been revised.
- Line 263: “n-fold method” is not how one would normally call in English a computation of the mean error from a sample of size n.
Answer: Thank you for your correction. the error description has been deleted.
- Line 294: the drying coefficients are not of much interest without knowing more about the fuel beds, such as the distribution of the particle sizes and shape, and the packing ratio. That should be defined and specified here.
Answer: Thank you for your advice. Adding the range of bed compactness in the manuscript.
- Line 305: Define fuel bed compactness and provide a reference. Packing ratio is normally used to measure compactness, which is a qualitative term, not quantitative.
Answer: Thank you for your advice. The definition of compactness has been added in Section 2.1. The compactness is the same as that of packing ratio. (Bradshaw L S, Deeming J E, Burgan R E, et al. (1983). The 1978 National Fire-Danger Rating System: Technical Documentation. USDA Forest Service, Intermountain Forest and Range Experiment Station Ogden, Utah 84401, General Technical Report INT-169. )
- Line 306: Define how is the “influence” computed and provide a reference. Maybe some kind of influence coefficients? “influence” is used elsewhere in the paper in the common English sense. Confusing.
Answer: The expression here is not precise, and the full manuscript is changed to effect.
- Line 353: Numbering of the lines ends. Please number the lines all the way to the end.
Answer: Thank you for your advice, It has been added.
- Section 5.1, comparing the drying coefficients: the results are about twice less than the cited one. This is not “slightly”.
Answer: Thank you for your advice. The “slightly” has been deleted.
